# A New Study on Fushi of Early Quanzhen Daoism

**Hongyi Chen and Yongfeng Huang \***

Department of Philosophy, Xiamen University, Xiamen 361005, China; 10420200156437@stu.xmu.edu.cn
**\*** Correspondence: yongfeng_huang1976@163.com

**Abstract:** Fushi (服食), a method for treating diseases and nourishing life to achieve longevity, is highly valued and widely used in traditional Taoism. Regarding whether Quanzhen Taoism, a new form of Taoism founded in the Jin Dynasty (1115–1234), practices Fushi, contradictory opinions have been recorded in *Collected records written on Qingyan Mountain* (Qingyan conglu 青巖叢錄) from the end of the Yuan Dynasty (1271–1368) and *The History of the Taoist School founded by (Qiu) Changchun* (Changchun daojiao yuanliu 長春道教源流) from the late Qing Dynasty (1636–1912). Today's scholars generally believe that Quanzhen Taoism emphasizes the cultivation of heart and mind and thus has nothing to do with Fushi. This article, centered around early Quanzhen Taoism representatives Wang Chongyang 王重陽 (1112–1170) and the "Seven True Ones (Qizhen 七真)", combines their writings, quotations, biographies, and other materials and discovers that while Wang Chongyang and others heavily criticized the traditional method of Fushi, they also carried out extensive Fushi activities and accumulated rich practical experience in areas such as taking medicine (fuyao 服藥), breathing exercises (fuqi 服氣), fasting (bigu 辟穀), dieting (yinshi 飲食), and using talismans (fufu 服符). Early Quanzhen Taoism both denied and utilized Fushi leading to a contradiction between words and deeds. The reasons for this contradiction can be attributed to two aspects: the internal alchemy thinking of the early Quanzhen Taoism that prioritized Tao over technique (shu 術), and dual cultivation of inner nature (xing 性) and life (ming 命) and prioritizing the former over the latter.

**Keywords:** Fushi; Quanzhen Daoism; Wang Chongyang; Qizhen (Seven True Ones)

## 1. Introduction

Fushi is one of the common practices of physical cultivation in Daoism, which aims to heal diseases, strengthen the body, prolong life, and even achieve immortality through the intake of specific drugs, food, gas, talisman water, etc. As early as the Spring and Autumn (770–476 BC) and Warring States (475–221 BC) periods, Fushi was popularized among immortals and alchemists. After the birth of Daoism at the end of the Eastern Han Dynasty (25–220), this practice was absorbed and transformed, gradually developing into a rich and diverse set of methods for nourishing the body through Fushi. Many scholars have already conducted research on this Taoist phenomenon, accumulating quite substantial results.[1]

Among the many sects of Taoism, the Fushi practices of Quanzhen Taoism are especially noteworthy. Quanzhen Taoism arose in the middle of the Jin Dynasty, founded by Wang Chongyang in northern China. It broadly absorbed the intellectual essence of Confucianism, Buddhism, and Taoism and pursued the goal of achieving immortality, advocated the dual cultivation of inner nature and life (xingming shuangxiu 性命雙修) and the equal emphasis of accomplishments and practices (gongxing bingzhong 功行並重). It innovated comprehensively the Taoist doctrines, commandments, temple systems, methods of practice, and transcendent states. Therefore, Chen Yuan 陳垣 (1880–1971) summarized the ideological features of Quanzhen Taoism with the phrase "neither Confucian, Taoist nor Buddhist", referring to it as the "New Taoism" or the "reformist sect within Taoism" (Chen 1962, p. 2). Influenced by this view, as a late-emerging sect, Quanzhen Taoism is often emphasized by its researchers in terms of the new aspect. According to existing

Quanzhen Taoist material, members of Quanzhen Taoism frequently elucidate the meaning and methods of mental cultivation. In contrast, they rarely discuss physical cultivation. Even if they do touch upon it, it is usually related to internal alchemy and there is almost no content specifically focused on Fushi. Therefore, scholars studying Quanzhen Taoism, when investigating its religious practices, seldom pay attention to the relationship between Quanzhen Taoism and Fushi practices, and even directly assert that Quanzhen Taoism has nothing to do with Fushi. Other treatises focusing on Taoist Fushi practices usually consider Taoism as a whole, based on Taoist texts specifically discussing Fushi. Obviously, this method is not applicable to Quanzhen Taoism, which has not left any writings on Fushi, so these works naturally do not pay attention to the Fushi situation of Quanzhen Taoism. If we only focus on the features of Quanzhen Taoism that are different from traditional Taoism and ignore the usage of traditional Taoist methods by Quanzhen Taoism, we cannot fully and accurately grasp the content and social activities of Quanzhen Taoism's practices, and our understanding of Quanzhen Taoism's inheritance in the history of Taoist development will be insufficient. Therefore, it is very necessary to explore the relationship between Quanzhen Taoism and Fushi practices, especially the relationship between them during its initial period of establishment.

The representative figures of early Quanzhen Taoism mainly include the founder Wang Chongyang and his seven disciples known as the "Seven True Ones": Ma Yu 馬鈺 (1123–1183), Tan Chuduan 譚處端 (1123–1185), Liu Chuxuan 劉處玄 (1147–1203), Qiu Chuji 丘處機 (1148–1227), Wang Chuyi 王處一 (1142–1217), Hao Datong 郝大通 (1140–1212), and Sun Bu'er 孫不二 (1119–1182). The relationship between early Quanzhen Taoism and *Fushi*, which this article discusses, revolves around Wang Chongyang and the "Seven True Ones". Since Quanzhen Taoism has not discussed *Fushi* in the form of a monograph or a special chapter, we cannot use the traditional method of examining Taoist *Fushi* practices. Instead, we can only compile trivial segments about *Fushi* from the materials of early Quanzhen Taoism. We mainly selected materials from the following channel. First, the poetry collections and quotations of Wang Chongyang and others, which were compiled by the disciples of Wang Chongyang and the "Seven True Ones", are collected. These recorded the educational and cultivated thoughts of Wang Chongyang and others in the form of poetry and recorded sayings. Second, the Taoist books of Quanzhen Taoism written by Taoist priests are selected. These recorded the life stories of Wang Chongyang and others in the form of individual biography. A considerable part of these are stele inscriptions with clear chronological records and are highly credible. Third, other documents from the Jin and Yuan periods, which were created at times relatively close to the active periods of Wang Chongyang and others, are collected. Therefore, they can more realistically reflect the situation of early Quanzhen Taoism. The *Fushi* practices under investigation include taking medicine, absorbing energy, grain avoidance, diet, and taking talisman water.[2]

## 2. Early Quanzhen's View on Fushi

Regarding the relationship between Quanzhen and Fushi, there have been different opinions since the Ming and Qing Dynasties. Wang Hui 王禕 (1322–1373) in *Collected records written on Qingyan Mountain* states: "However, there are two techniques in the art of immortals: one is cultivation, and the other is Fushi. These two are now the teachings of Quanzhen" (Wang 2016, p. 607). He also says: "Nowadays, cultivation and Fushi techniques have been passed down and are used by Quanzhen. The name of Quanzhen rose during the Jin Dynasty, with divisions between the Southern and Northern sects. The Southern Sect emphasizes the nature, while the Northern Sect emphasizes life" (Wang 2016, p. 608). Wang Hui learned the Confucian classics from Huang Jin 黃溍 (1277–1357), and was a schoolmate of Song Lian 宋濂 (1310–1381). From 1347 to 1349, Wang Hui compiled historical books in Yanjing 燕京 (now Nanjing 南京, Jiangsu 江蘇), during which he submitted letters to the authorities, expressing his views on the social situation, but was not taken seriously. Therefore, in 1350, Wang Hui left Yanjing, retired to Qingyan Mountain 青巖山 (now Yiwu 義烏, Zhejiang 浙江), and wrote the book *Collected records*

*written on Qingyan Mountain*. In 1358, Wang Hui entered the service of Zhu Yuanzhang 's 朱元璋 (1328–1398) regime. In 1369, Zhu Yuanzhang issued a decree, ordering Song Lian and Wang Hui to compile *Yuan History* (Yuanshi 元史). In 1373, Wang Hui was sent on a mission to Yunnan 雲南 by imperial order and was killed. *Collected records written on Qingyan Mountain* is Wang Hui's reading notes, which narrate the origins of the *Book of Changes* (Zhouyi 周易), *Book of Documents* (Shangshu 尚書), *Book of Odes* (Shijing 詩經), *Spring and Autumn Annals* (Chunqiu 春秋), Three Rites (Sanli 三禮), *Centrality and Commonality* (Zhongyong中庸), *Great Learning* (Daxue 大學), Apocrypha, Buddhism, Taoism, Geomancy, Medical school, etc. When Wang Hui narrates the origins of Taoism, he starts from Laozi's 老子 *Scripture of the Dao and Its Virtue* (Daode jing 道德經) and goes all the way to Zhengyi 正一 and Quanzhen Taoism at his time. Although Wang Hui was greatly influenced by Confucian thought, he also had many interactions with Taoist priests. His division of the Taoist pattern at that time into Zhengyi and Quanzhen schools indicates that he accurately grasped the basic situation of Taoism at that time. This indicates that in Wang Yi's time, the emphasis on Fushi and cultivation were the two main features of Quanzhen. Around the middle of the Yuan Dynasty (1271–1368), the Southern Sect led by Bai Yuchan 白玉蟾 (1134–1229) and the Northern Sect led by Wang Chongyang merged to form a larger Quanzhen. In terms of organizational form, the Northern Sect unified the Southern Sect; in terms of cultivation methods, the Southern Sect influenced the Northern Sect. The Southern Sect emphasized physical cultivation and widely used various techniques for conserving essence and nourishing qi (炁). Therefore, some scholars believe that "Wang Yi directly regards Quanzhen as a teaching of cultivation and Fushi, which cannot be denied as a direct result of the blending and confluence of Daoist sects since the middle and later periods of the Yuan Dynasty" (Zhu 2009, p. 96). However, *The History of the Taoist School founded by (Qiu) Changchun*, completed in the year of 1879, presents an opposing view: "The teachings of Qiu Chuji has gained much from the essentials of *Scripture of the Dao and Its Virtue* and has no drawbacks from the inferior practices of cultivation, Fushi, talismans, exorcisms, or rituals" (ZW 949, 1). The author Chen Jiaoyou 陳教友 (1824–1881), was a seventeenth-generation disciple of Quanzhen in the late Qing Dynasty. He believed that Qiu Chuji (also known as Changchunzi 長春子) focused on inheriting and promoting Laozi's knowledge, with no connection to Fushi or other Daoist techniques. Based on this, Chen Jiaoyou extended this judgment to the whole Quanzhen: "In the early Ming Dynasty, Daoism flourished with Quanzhen and Zhengyi . . . Quanzhen originally did not mention cultivation or Fushi. Since Chen Zhixu's 陳致虛 (1290-?) *Folios on Awakening to Perfection* (Wuzhen pian 悟真篇) studies, the Southern and Northern sects have been unified, and only then did cultivation begin, but Fushi has not been heard of" (ZW 949, 147). In his view, Quanzhen has had no connection with Fushi from beginning to end. It can be seen that the two have opposite views on whether or not there was the taking of medicine in Quanzhen Daoism, but both advocate that there was no such practice in early Quanzhen Daoism. This understanding has great influence and has been supported by many scholars in later generations.[3]

Based on the available materials, Wang Chongyang and others basically rejected and negated the Fushi techniques of traditional Daoism. Wang Chongyang said: "Do not rely on scattered pills to seek virtue, nor depend on talisman water to hope for ascension" (DZ 1153, 9. 6b). He also said: "Having obtained the carefree true freedom, why bother with taking pills or dining on mist?" (DZ 1153, 10. 10a). Things such as talisman water, taking pills, and dining on mist are of no help in accumulating virtue and transcending the world. Only the carefree state of mind is the path to liberation. Ma Yu also said: "Practicing meditation, paying respects to the earth, chanting, dining on mist, and fasting while reading scriptures. Relying on talisman water and concentrating on imagination, gargling and swallowing exhaust the body. Many are lost in the arts of the bedroom, depending on Fu Huan Yuan (Returning to the Origin), and water and fire as support. If not sinful, various practices are obstructed, and cultivation is misguided" (DZ 1150, 8a). He also said: "Stop talking about and swallowing, don't mention concentration, swallowing mist and

taking Qi are all illusory. Sitting in meditation and practicing diligently, pulling hands and legs is laborious. Picking and fighting for divine elixir is scattered and lost, taking pills, and water and fire are inappropriate. In the end, it is the principle of Wu Wei (non-action) that is most reliable" (DZ 1150, 20b). The mentioned practices such as dining on mist, swallowing mist, fasting, talisman water, and taking Qi are all essential aspects of Daoist Fushi. Ma Yu pointed out that only the ultimate principle of nature and non-action is worth relying on, and the practices of Fushi are all in vain. *Record of the Celebrated Encounter of the Arcane Wind* (Xuanfeng qinghui lu 玄風慶會錄), completed in 1232, recorded Qiu Chuji's answer to Genghis Khan's 成吉思汗 (1162–1227) question about the path to longevity: "Ancient people said: 'Taking medicine for a thousand mornings is not as good as sleeping alone for one night.' Medicine is made of plants, and energy is the marrow. So what is the benefit to remove the marrow and add the plants? For example, if you store gold in a bag, and gradually remove the gold and add iron, over time the gold will be gone and the bag will be filled with iron only. What's the difference between the principle of taking medicine and this?" (DZ 176, 5b). The "medicine" mentioned here refers specifically to herbal medicine. Qiu Chuji believed that the vital substances ("essence") in the human body are the essence of a person, while medicines are merely various plants. If a person does not conserve their own essence but instead pursues taking herbal medicine, it is not different from neglecting the fundamentals and pursuing the superficial. It can be seen that Wang Chongyang, Ma Yu, and Qiu Chuji all held a negative and critical attitude towards the traditional Fushi techniques of Daoism.

### 3. Fushi Practices in Early Quanzhen Daoism

Since Taoism incorporated *Fushi* practices into its cultivation system, it has developed a rich variety of *Fushi* programs. Previous scholars have summarized Taoist *Fushi* practices into five types: taking medicine, absorbing *qi*, grain avoidance, diet, and taking talisman mater, which basically cover all Taoist *Fushi* activities. Therefore, we will start from these five aspects to examine the *Fushi* practices of early Quanzhen Taoism.

Although early Quanzhen Daoism did not advocate the traditional Daoist practice of using medicinal herbs, they did not entirely reject it in practice. Wang Chuyi, one of the "Seven True Ones", often consumed herbs while living in seclusion on Tiezha Mountain in Wendeng County (now Weihai 威海, Shandong 山東). *Song of the Stone Fungus on Tiezha Mountain* (Yong chashan shizhi 詠查山石芝) says: "The essence of the sun and the moon form auspicious sprouts, the mysterious light and true qi are contained within. It nourishes the five viscera, producing golden fluid, and secretly strengthens the entire body's muscles and bones" (DZ 1152, 1. 30a). Another *Song of the Stone Fungus on Tiezha Mountain* (Yong tiechashan shizhi 詠鐵查山石芝) states: "Among spiritually beautiful mountains and seas, to pick fungus, one must forget their form so as to nourish the jade sprouts to satisfy hunger and quench thirst, increase spirit and intelligence, and shine with clarity. People who take it will not decline or decay" (DZ 1152, 4. 21a-b). The "stone fungus" (shizhi 石芝) mentioned here refers to the fungus that grows on stones, which has long been included in the list of consumables by Daoists.[4] According to Wang Chuyi, stone fungus has the effects of cultivating true qi, nourishing the five viscera, strengthening the body, and prolonging life. In addition, Wang Chuyi often exchanged medicines with others. *Thanking county magistrate for Mint Pills from Yunzhou* (Xie zaigong hui yunzhou bohe jianwan 謝宰公惠鄆州薄荷煎丸) says: "A drop of the elixir can nourish a beautiful complexion for a thousand generations and people break through the cycle of life and death" (DZ 1152, 2. 28a-b). "*Seeking Atractylodes from the Same Province to Give* (Benzhou tongzhi mi cangshu zengzhi 本州同知覓蒼術贈之)" says: "The light of the sun and the moon form a beautiful knot, gradually making the muscles and bones strong. It can gather the divine light of the four seasons, and transform into immortality" (DZ 1152, 2. 29a-b). The mint pills and Atractylodes mentioned in the poems were originally medicines for treating diseases, but Wang Chuyi's focus was not on their healing functions but on their magical effects in promoting cultivation. Qiu Chuji had many connections with technique of Tak-

ing Drugs.[5] In his *Demonstration to the Public* (Shizhong示眾), he wrote: "Stone marrow can prolong life, and cinnabar can rejuvenate the complexion. Ge Hong 葛洪 traveled the great sea, and Wang Lie 王烈 (supposedly circa 224–263) encountered deep mountains" (DZ 1159, 4. 11b-12a). The "stone marrow" (shisui 石髓) and "cinnabar" (dansha 丹砂) mentioned in the poem are all external alchemy medicines, and both Ge Hong and Wang Lie are representative figures of medicinal cultivation. It is obvious that Qiu Chuji also had an affirmative attitude to the method of seeking longevity through ingesting external elixirs. In addition, in his *Discourse on Conserving Health* (Shesheng xiaoxi lun 攝生消息論), he associated spring, summer, autumn, and winter with the liver, heart, lung, and kidney of the human body, respectively, reminding people to pay attention to the corresponding visceral diseases in different seasons and listing the medicinal recipes for treating the diseases. In summary, early Quanzhen Daoism not only understood and valued the traditional Daoist practice of using medicinal herbs, but to some extent still retained the legacy of medicinal cultivation.

Among the five types of dietary practices in early Quanzhen Daoism, the most commonly used method was taking qi. Early Quanzhen Daoism's practice of taking qi usually took place within the framework of internal alchemy. The ingested qi included both external and internal qi, with more emphasis on internal qi. Sun Bu'er's *Kun Dao Kung Fu Sequence* (Kundao gongfu cidi 坤道功夫次第) in the tenth chapter titled *Dietetic Regimen (fushi 服食)* states: "Great alchemy forms in the mountains and marshes, containing the essence of creation. In the morning, we embrace the qi of the sun, and at night, we absorb the essence of the moon. When the time is right, the elixir can be collected, and as years pass, the body becomes lighter. When the original spirit can move freely, all pores emit light" (JY 177, 444). Practitioners in the mountains and marshes absorbed the qi of the sun and the moon during the early morning and evening. After persisting for a period of time, the elixir naturally forms within their bodies, making them light and agile. When the original spirit can move freely, light will emanate from all pores. This is how early Quanzhen Daoists applied the technique of ingesting external qi to the female elixir cultivation method. In terms of ingesting internal qi, Wang Chongyang and others had accumulated a wealth of experience, mainly including "Drinking Daogui" (yin daogui 飲刀圭) and "Eating Shu Rice" (can shumi 餐黍米). The term "Daogui" frequently appears in the poetry of Wang Chongyang and the "Seven True Ones". Wang Chongyang says, "Frequently drink Daogui" (DZ 1153, 12. 19a-b). Ma Yu says, "Feast daily on Yurui (jade stamens), and drink Daogui often" (DZ 1142, 1. 25b). Liu Chuxuan says, "Leisurely drink Daogui throughout the twelve zodiac signs" (DZ 1141, 3. 9a). Tan Chuduan says, "Always be open and honest, and silently drink Daogui" (DZ 1160, 2. 7b). Examples such as these are numerous. "Daogui" originally referred to the tools used to measure medicine in external alchemy and was later introduced into internal alchemy practice. Wang Chongyang explained, "Drinking Daogui is swallowing saliva to ingest qi" (DZ 1156, 10a). It can be seen that "Drinking Daogui" is the traditional Daoist method of swallowing saliva to ingest qi. He says, "Daogui is one. With water and qi, it can give birth to everything" (DZ 1156, 16a-b). He also says, "Exhaling is the Dao, inhaling is the Gui, practicing is everything, and the verifier is the Daogui, which is the rejuvenation" (DZ 1156, 16b). This shows that the qi being ingested is the vital qi cultivated and nourished within the body. The term "Shu Rice" was also frequently mentioned by Wang Chongyang and others. Wang Chongyang says, "Before the Three Pure Ones, participate in the Shu Rice Pearl" (DZ 1153, 12. 9a). Ma Yu says, "In the beginning, the elixir forms a mysterious treasure, and the Shu Rice Pearl hides the external spring" (DZ 1149, 3. 10b). Sun Bu'er says, "Every day, increase the Shu Rice, and the white hair will turn red again" (JY 177, 444). In Qiu Chuji's *Straightforward Directions on the Great Elixir* (Dadan zhizhi 大丹直指), he specifically explained: "The dragon is the positive yang qi of the heart fluid, and if it doesn't go up, it naturally combines with the kidney qi. The tiger is the true water of the kidney qi, and if it doesn't go down, it naturally intersects with the heart fluid. When the dragon and the tiger mate, a grain shaped like Shu Rice is formed. This method is called the dragon and tiger mating, and it reveals



the elixir" (DZ 244, 1. 6a). He also says, "This is the combination of qi, called mating. Each day, perform one mating, obtain one object shaped like Shu Rice, and return it to the Yellow Court in the center, which can naturally increase longevity and prolong life" (DZ 244, 1. 7b). From this, it can be seen that the so-called "Shu Rice" is actually the product formed by the combination of the qi of the heart and kidneys. In terms of its nature, it also belongs to internal qi.

Early Quanzhen Daoists also paid attention to abstaining from grains in their cultivation practices. Wang Chongyang, in *Su Muzhe—Persuading the Same Stream* (Su Muzhe—Quan tongliu 蘇幕遮·勸同流), said, "The residue of the five grains is difficult to transform" (DZ 1153, 4. 9b). This can be seen as a direct continuation of the traditional Daoist practice of abstaining from grains. He was also influenced by Buddhist views on the body, repeatedly mentioning "four false bodies" (sijia shenqu 四假身軀), "four false mortal bodies" (sijia fanqu 四假凡軀), and "illusory transformed color bodies" (huanhua seshen 幻化色身), expressing a denigration and negation of the physical body. Because "the flesh and the four elements are false", and eating grains is for the maintenance of the body, he reminded others that "you should know that the grains support the body", comparing eating grains to "eating dirt, dining on mud, and nourishing dust" (DZ 1153, 2. 11a). However, he opposed abstaining from grains intentionally, believing that it should be a natural result of internal alchemy cultivation. In *Requesting the Master to Abstain from the Five Grains since Childhood* (Yugong qiu ziyou bushi wugu 于公求自幼不食五穀), he wrote: "This cause is only in Yujing Mountain, no need for feasting and paying attention to appearance. Nourishing qi relies on the moisture of true water, while nourishing the spirit is like the leisure of white clouds. When practicing to the point of overflowing and transcending the middle position, lead and mercury are cast out from here. Just wait for the impurities outside to be exhausted, then the brilliant colors will return" (DZ 1153, 1. 20b–21a). When discussing Cutting Off Grains, Wang Chongyang used terms such as "True Water" (zhenshui 真水) and "Lead and Mercury" (qiangong 鉛汞) to advocate inner alchemy-oriented practice of it. According to Wang Chongyang, abstaining from grains is not an independent cultivation technique but a characteristic that arises when internal alchemy cultivation reaches a certain level.[6]

Sun Bu'er's *"Kun Dao Kung Fu Sequence"* in Eleventh Song *Abstaining from Grains* (Bigu 辟穀) also says: "Once you have obtained the spiritual qi, the lungs and viscera will feel cool and extraordinary. Forget the spirit without attachment, and unite with the extreme emptiness. In the morning, you eat mountain yams; when hungry at dusk, gather marsh mushrooms. If you mix with smoke and fire, the body will not tread the jade pond" (JY 177, 444). In Sun Bu'er's female alchemy practice, when the practitioner's qi is full, they will feel their organs refreshed, indescribable, and enter a carefree state of "forgetting the spirit " (wangshen 忘神) and "uniting with the extreme" (heji 合極). At this point, practitioners can naturally abstain from grains and must be careful not to eat them again, replacing them with high-nutrition and hard-to-digest foods such as mountain yams and lotus seeds.[7] From this, it can be seen that both Wang and Sun advocated achieving a natural state of abstaining from grains through internal alchemy cultivation.

Early Quanzhen Daoism combined religious cultivation with daily diet. Firstly, Wang Chongyang and others advocated that diet should be simple and moderate. Wang Chongyang said: "People should restrain their desires, and their intake of sour and spicy food should be regulated" (DZ 1153, 11. 11b). This straightforwardly expresses the need to suppress the desires of the mouth and stomach. Ma Yu proposed "Ten Precepts", (shijie 十戒) the seventh of which states: "Be careful with your words, moderate your diet, reduce your taste, discard glory, and eliminate hatred and love" (DZ 1256, 1. 9a). He emphasized controlling the quality and quantity of food and drink. Tan Chuduan said: "When I feel cold and hungry, I only wear coarse clothes and eat pickles" (DZ 1160, 1. 5a). Liu Chuxuan said: "Do not think about delicious meals, just beg for leftovers" (DZ 1141, 3. 19a). Both advocated a frugal lifestyle that does not seek delicious food. Qiu Chuji said: "learning the Dao, monks wear coarse clothing, eat poor food, and do not accumu-

late wealth for fear of harming their bodies and losing their blessings. Learning the Dao, laymen should also moderate their food and drink, dwelling place, treasures, and wealth according to their situation, and should not be too different" (DZ 176, 7b-8a). The identities of the practitioners, whether they are monks or laymen, have different requirements, but the requirement to control their diet is the same. Secondly, Wang Chongyang and others, from the perspective of cultivation, repeatedly emphasized abstaining from alcohol and meat. Wang Chongyang said: "In general, to learn the Dao, one must not kill, steal, drink alcohol, eat meat, break precepts, or violate vows" (DZ 1154, 2. 3a). He clearly regarded alcohol and meat as obstacles to learning the Dao. Ma Yu also said: "Alcohol is the pulp that disrupts nature, and meat is the thing that cuts off life. It is best not to eat it" (DZ 1057, 2b). Quanzhen Daoism takes life cultivation as a means to become an immortal, and alcohol and meat are things that damage inner nature and life. He also praised his disciples, saying, "Each one is pure and clean for life, abstains from desires, meat and alcohol" (DZ 1057, 14b). Tan Chuduan said: "Everyone who learns the Dao craves for meat, and those who are addicted to alcohol and flowers are entangled in sins. Desiring meat hurts the lives, and the intention of greed and love harms the root" (DZ 1160, 3. 5a). Eating meat harms the lives of creatures, and drinking alcohol easily gives rise to a heart of greed and love. Both will lead people astray from the right path of learning the Dao. Liu Chuxuan said: "while eating meat, you should think of living animals, and you should not drink alcohol for no reason" (DZ 1141, 4. 15a). He also advised people to refrain from eating meat and drinking alcohol. Thirdly, early Quanzhen Daoism used tea drinking as an auxiliary means of cultivation. Ma Yu's *Dreamless Command—Following Chongyang's Rhyme* (Wumengling—Danyang jiyun 無夢令·丹陽繼韻) says: "No matter how many bowls Zhaozhou has, don't call Lu Tong anymore. Pray to the Duke of Taiyuan, free from the entanglement of sleep demons. Bright and splendid, bright and splendid, see the shore of eternal life" (DZ 1154, 1. 13a). The "Zhaozhou 趙州" mentioned here refers to the master Congshen 從諗禪師 (778–897), who was famous for his key phrase "Go and have tea"; the "Duke of Taiyuan 太原公" refers to Bai Juyi 白居易 (772–846), who, along with Lu Tong 盧仝 (769?–835), wrote poems about tea. Ma Yu pointed out that drinking tea could help avoid the intrusion of sleep demons, allowing one to enter a bright and splendid realm of achieving immortality. In *Walking on Clouds—Tea* (Tayunxing—Cha 踏雲行·茶), it is said: "Tea deserves the name as it eliminates the thoughts that disrupts sleep, and the merit is immeasurable. Confucius no longer dreams of Duke Zhou, the mountain people laugh at Chen Tuan's drowsiness. The seven bowls of Lu Tong, and the monk Zhaozhou, ever know that the taste returns to the supreme. If Zaiyu could taste a cup, there would be no daytime sleep, and the spirit would be refreshed" (DZ 1142, 1. 10a–b). Here, a series of allusions such as Confucius 孔子 (551–479 BC) not dreaming of Duke Zhou 周公 (supposedly circa 1096–1032? BC), Chen Tuan's 陳搏 (871–989) drowsiness, and Zaiyu's 宰予 (522–458 BC) daytime sleep have been used to illustrate that by drinking tea, can overcome sleep problems and achieve a refreshed and clear state. Early Quanzhen Daoism believed that during sleep, people cannot control their thoughts and easily generate desires, which hinders cultivation. Therefore, they adopted the method of "fighting sleep demons" (zhan shuimo 戰睡魔) to restrain the desire for sleep, reduce sleep time, and maintain inner purity. Modern scientific research shows that tea contains a large amount of caffeine and catechins, which can excite the central nervous system of the human body. Therefore, Ma Yu relied on drinking tea to dispel sleepiness and maintain a vigorous mental state. Finally, early Quanzhen Daoism proposed dietary principles that correspond to the changes of the four seasons. Qiu Chuji's *Discourse on Conserving Health* extensively absorbed and drew upon the principles of health preservation from before the Jin Dynasty. Based on the correspondence between human organs and the changes in the four seasons and the five elements, he proposed a set of general principles and specific methods for dietary health preservation in different seasons.[8] The content is concise and practical, making it suitable for both Quanzhen Taoists and the general public to follow.

Since the advent of Daoism, the practice of using talisman water to treat diseases has been passed down continuously and was later adopted by the Quanzhen Sect of Neo-Daoism. Wang Chongyang said: "How can you allow medical treatment? Do not let people be overwhelmed, the use of talisman water is a good cause" (DZ 1153, 7. 12b). This fully affirms the positive effect of practicing talisman water on accumulating good causes. In his work dedicated to immortal Lu 陸仙, he openly said, "I practice talisman water, and you cultivate medicinal herbs" (DZ 1154, 2. 3b). In *Walking on Grass—Kunyu Mountain Tuan'an* (Tasuoxing—Kunyushan tuan'an 踏莎行·昆崙山團庵), it is said, "Water talisman does not work here, the cause and effect have already reached Wendeng County" (DZ 1153, 7. 13b). The "water talisman" mentioned here is equivalent to "talisman water" (Kunio 2006, p. 351). Since Wang Chongyang said that he did not perform talisman water here, it implies that he had done so before. Moreover, Wang Chongyang also used talisman water to cure his disciple Ma Yu's illness. In the *Records of the Correct Lineage of the Golden Lotus* (Jinlian zhengzong ji 金蓮正宗記) written in 1241, it is recorded: "Then he went to the smoke and mist cave on Kunyu Mountain. The teacher suddenly suffered a severe headache as if his head was splitting. People said, 'Master Ma may not survive these days!' The real person (Wang Chongyang) said, 'I have transformed him from three thousand miles away. Can I let him die?' He then chanted water for him to drink, and he was healed after drinking" (DZ 173, 3. 5b). The "talisman water" mentioned here is the method of treating diseases with talisman water. Wang Chongyang cured Ma Yu's headache through this method. Ma Yu also affirmed and utilized talisman water. In his *Praising fashi Liu's Ascension with a Poem* (Hunyuan Liu fashi shenghua, yi ci zan zhi 混元劉法師昇化，以詞讚之), he highly praised Master Liu's 劉法師 actions of "writing talisman water to cure diseases and save disasters" (DZ 1149, 10. 23a). In *Presenting Ma Yan Gao* (Zeng Ma Yangao 贈馬彥高), he said, "When encountering me, you forget your worries by drinking talisman wine, which can cure hundreds of diseases like saving a drowning person" (DZ 1149, 9. 11a). The "talisman wine" (zhoujiu 咒酒) mentioned here should be similar to "talisman water", just using wine as a medium instead. In addition to "talisman water", Ma Yu also used a technique called "talisman fruit" (zhouguo 咒果). The *Records of the Correct Lineage of the Golden Lotus* records: "A poor man in Zhiyang had both of his feet disabled and cried out in pain. The teacher (Ma Yu) chanted water for him to drink, and he walked as if flying. Luan Wugong 欒武功 (fl. circa 11th c), who had suffered from wind paralysis for a long time, found no effect from hundreds of medicines. The teacher (Ma Yu) chanted fruit for him to eat, and he was healed in one day" (DZ 173, 3. 10a). Whether it was "talisman water" or "talisman fruit", the therapeutic effects were astonishing, and the time it took to see results was extremely short. Due to a lack of historical materials, we cannot know the specific operation of "talisman fruit". Judging from the name alone, it may involve drawing talismans directly on the fruit or using fingers to draw talismans on the fruit's surface, then consuming it. In essence, it should be an expanded application of the talisman water method.

## 4. The Internal Causes of Inconsistency in Fushi in Early Quanzhen Daoism

In the early days of Quanzhen Daoism, practitioners used Fushi techniques for cultivation and preaching while simultaneously belittling and rejecting it. The appearance of this contradictory situation between concept and practice is not only related to the influence of the Zhong-Lü 鍾呂 thought on early Quanzhen Daoism, but also due to the early Quanzhen Daoist tendency to prioritize spiritual cultivation over preserving life.

### 4.1. Emphasizing "Dao" over "Technique" and the Inner Alchemy Thinking of Cultivation Methods Leading to the Dao

Taoism, named after "Dao", is known for its diverse methods. Therefore, the pair of concepts, "Dao" and "Technique", are widely applied in the categories of Taoist cultivation and transcendence. Generally speaking, in Taoism, "Dao" has two meanings. On one hand, it refers to the origin and essence of all things in the universe. From the perspective of cosmic origin, "Dao" is the beginning of heaven and earth (the world). For all things

themselves, everything in the world depends on Dao for their creation. It exists eternally and is the ultimate destination of the world and all things, transcending the tangible material world. On the other hand, "Dao" also refers to the realm of self-transcendence of life. Through cultivation, humans can approach and return to "Dao", which can bring both body and spirit to the highest level of perfection. Technique refers to all kinds of concrete and practical methods of cultivation, all aiming to approach Dao. Taoism believes that "Dao" and "Technique" complement each other. "Dao" provides a rich theory for "Technique", pointing out the direction for the development of "Technique", while "Technique" is the prerequisite and path to achieving "Dao". Since the concepts of "Dao" and "Technique" can describe the cultivation activities of Taoism, they can naturally be applied to the Inner Alchemy technique, which has dominated Taoist cultivation since the end of the Tang Dynasty (618–907) and the Five Dynasties (907–979). Taoist Inner Alchemy theory believes that the Inner Alchemy practice follows the "Dao" of creating heaven and earth and generating all things. Carrying out practical cultivation based on it can lead the practitioner to the "Dao" of life transcendence. As a new Daoist school based on Inner Alchemy cultivation, early Quanzhen Daoism often emphasized the Dao and techniques simultaneously, showing its preference for the Inner Alchemy Dao and not engaging in other cultivation methods. Wang Chongyang, in his *Ninghai Begging for Change on Paper Banners* (Ninghai qihua shu zhiqishang 寧海乞化書紙旗上), said, "People ask for evidence against the harmful wind, but I have no techniques. I only eat and sleep, following the natural human instincts, which is full of the spirit of the great Dao of non-action, corresponding to the absence of techniques" (DZ 1153, 2. 18a). The *Three States and Five Associations Begging for Merit* (Sanzhou wuhui huayuanbang 三州五會化緣榜) further states, "You do not understand the root cause and only learn superficial techniques, which only helps you to seek blessings and nourishing the body, not cultivate life and enter the Dao" (DZ 1154, 3. 13a). In the *Song of exhortation to the Dao* (Quandao ge 勸道歌), Wang Chongyang listed various traditional Daoist techniques, such as fasting, meditation, talisman water, and gargling, and sternly pointed out, "Do not use any of these, as they are all best discarded", then turned the focus to Inner Alchemy cultivation (DZ 1153, 9. 4b–5a). Obviously, the "exhortation to the Dao" was actually encouraging people to abandon techniques and return to the Dao through Inner Alchemy cultivation. Ma Yu said, "Purity lies in purifying the heart and cleansing the qi. When the mind is clear, external things cannot disturb, so emotions settle and divine clarity arises; when the qi is pure, evil desires cannot interfere, so essence becomes complete and the abdomen is full. Hence, purifying the heart is like purifying water and nourishing qi is like raising a child. When the qi is refined, it becomes divine, and when the spirit is divine, the qi transforms, which is the result of purity. If one practices techniques with intention and action, they are just limited techniques; if one practices the principle of non-intention and non-action, it leads to the boundless realm of purity and emptiness" (DZ 1057, 8a–b). Ma Yu's "purity" (qingjing 清淨) cultivation discards limited techniques, advocating for the principle of non-intention and non-action. He believed that this can lead one back to the infinite and pure Dao. In the *Gift to Ju Deyi* (Zeng Ju Deyi 贈鞠得一), it is said that cultivation techniques are not the right path to return to the Dao; only by revealing one's true heart can one unite with the Dao (DZ 1149, 8. 16b). Tan Chuduan criticized those who only studied elixir techniques to show off their abilities but ignored the guidance of enlightenment and immortality (DZ 1160, 3. 5a–b). He also called for practitioners to recognize and understand the emptiness of all things, the concept of Buddhism, and let go of worldly thoughts and illusions, and learn the great Dao of immortality. Qiu Chuji said, "People in the world who pray for longevity do not make great vows from their original life and spirit but beg for blessings from immortals and Buddhas, which is abandoning the root and seeking the end. Our school does not talk about longevity because it transcends it. This supreme great Dao is not a trivial technique for extending life" (ZW 378, 284). In summary, there is a clear emphasis on the Dao over techniques in early Quanzhen Daoism literature. Therefore, when early Quanzhen Dao-

ism promoted Inner Alchemy and cultivation of the mind, it inevitably downplayed and disparaged techniques such as Fushi.

Wang Chongyang and others' understanding of "Dao" and "Shu" is actually inherited from the Zhong-Lü Daoist school. The early Quanzhen School revered Zhongli Quan 鍾離權 (fl. circa 9th c) and Lü Dongbin 呂洞賓 (fl. circa 9th c) as their ancestors, and absorbed and borrowed a large amount of Zhong-Lü teachings. In *Anthology of Zhongli Quan's Transmission of the Dao to Lü Dongbin* (Zhong-Lü chuandao ji 鍾呂傳道集), Zhongli Quan repeatedly criticized those who were obsessed with techniques and did not comprehend the Dao. He proposed the classification of cultivation into five levels of immortality and three levels of techniques, further explaining: "The three levels of techniques refer to the minor, intermediate, and major achievements. The five levels of immortality refer to ghost immortals, human immortals, earth immortals, spirit immortals, and celestial immortals, all of which are immortals" (DZ 263, 14. 2b–3a). In his description of human immortals, he said: "Human immortals are the second lowest among the five immortals. Cultivators who do not understand the great Dao, acquire one technique within the Dao, and have unwavering faith and determination throughout their lives. And The *qi* of the five elements interacts and consolidates, making them immune to the harm of the eight evil diseases, and they enjoy good health with few illnesses, which is called human immortality" (DZ 263, 14. 3b). Cultivators who cannot comprehend the Dao and stick to techniques will become human immortals. Although human immortals can manipulate the qi of the five elements to strengthen their bodies and reduce illness, this combination of qi is just a result of coincidences. Moreover, human immortals are helpless against aging, sickness, death, and suffering. However, this trend of praising minor techniques and ignoring the Dao spread widely in the secular society, which aroused Zhongli Quan's dissatisfaction. He said: "Techniques that do not conform to the Dao rely on extensive knowledge and strong recognition to deceive the world, praise each other and cause the great Dao to be unheard, thus giving birth to various minor techniques and side doors" (DZ 263, 14. 6a). He also said: "Since side door techniques are easy to achieve and popular among the secular people, they pass them on to each other and do not awaken until death. As a result, they become customs and corrupt the great Dao" (DZ 263, 14. 6b–7a). At the same time, Zhongli Quan also advocated that techniques should not be separated from the Dao. He said: "There are countless techniques, but they are all part of the Dao. If one cannot achieve the great Dao, one should stop at one technique within the great Dao. Once successful, one will live a peaceful and happy life with longevity, hence called human immortality" (DZ 263, 14. 4a). Although he did not highly regard the realm of human immortals, he still recognized that many techniques practiced to become human immortals were part of the great Dao. Apparently, Zhongli Quan's intention of belittling techniques was not to oppose them but to warn cultivators that these techniques were only aspects of the Dao, which were "lower techniques for nourishing life" and "minor techniques for gathering spirit". One should not be obsessed with or exaggerate them, otherwise, they would never truly grasp the great Dao and could only achieve human immortality in their lifetime. Zhongli Quan also showed cultivators a path to seek the Dao and immortality: "Minor techniques make human immortals, intermediate techniques lead to earth immortals, and Dao brings spirit immortals. These are the three levels of achievement, which are essentially the same. Seeking the Dao through techniques is not difficult; seeking immortality through the Dao is also quite easy" (DZ 263, 14. 5b). General techniques can only provide safety and longevity, belonging to the "minor achievements" level, but in terms of the goal of achieving immortality, they are consistent with the intermediate and major levels of cultivation, with only differences in the degree of attaining the Dao and the rank of immortality. Human immortals achieved through general techniques may not be praiseworthy, but they still stand at the starting point of cultivation, laying the foundation for further cultivation to become earth immortals and spirit immortals. This practice of integrating traditional Daoist techniques into the cultivation hierarchy was further summarized by Shi Jianwu 施肩吾 (fl. 820), who inherited the Zhong-Lü alchemy methods,

as the proposition of "cultivating techniques to enter the Dao" (lianfa rudao 煉法入道), becoming the bridge connecting the "old techniques" and the "new Dao".[9]

The formation of the Zhong-Lü Inner Alchemy School occurred during a critical period when the Inner Alchemy Dao was transitioning from hidden to prominent. On one hand, Inner Alchemy practitioners needed to attack the previous Daoist cultivation techniques to bolster the momentum of the emerging Inner Alchemy methods, highlighting their independence and advanced nature. On the other hand, various traditional Daoist cultivation techniques had a solid follower base for a long time before the maturity of Inner Alchemy Dao and had produced practical effects to some extent. Therefore, Inner Alchemy practitioners actively reconciled the relationship between the two and absorbed the achievements of relevant techniques to construct a more complete Inner Alchemy cultivation system. It is precisely because of the above reasons that Zhongli Quan adopted a more tolerant attitude towards the ingestion of herbal medicines and the cultivation of External Alchemy.[10] The early Quanzhen School's contradictory words and actions regarding the Fushi were one of the specific manifestations of inheriting the Zhong-Lü ideology. However, compared to Zhong and Lü, Wang Chongyang and others had greatly reduced tolerance to traditional Daoist techniques.

### 4.2. Dual Cultivation of Inner Nature and Life and Prioritizing the Former over the Latter

In its early stage, the Quanzhen School adhered to the principle of "dual cultivation of inner nature and life" in the Inner Alchemy tradition. Wang Chongyang said, "Life and nature are complementary" (ZW 866, 670). He also said, "Those who understand inner nature and life are truly practicing the Dao" (DZ 1156, 2a). It is evident that he regarded the two as a complementary and unified entity, and the realization of them is the right path to cultivation. Ma Yu said, "With a clear life, one attains longevity; with a tranquil nature, one can see far. Life is the name of qi, and nature is the word of the divine. qi is the mother of the divine, and the divine is the child of qi. When the child and mother become the true unity, one can transcend life and death" (DZ 1149, 5. 6b). In his view, only through the dual cultivation of inner nature and life, can one achieve true nature and transcend life and death. Tan Chuduan said, "when you let go of everything, inner nature and life can be harmoniously integrated" (DZ 1160, 2. 8a). This expresses the pursuit of harmony and completeness in inner nature and life. Liu Chuxuan said, "Without water at its root, the sprout dies; without life for its nature, the body dies" (DZ 1058, 4b). He used the analogy of root and sprout to illustrate the inseparable relationship between essence and life. Qiu Chuji also said, "The secret of the Golden Elixir (jindan 金丹) lies in one essence and one life only. Essence is heaven, always hidden at the top; life is earth, always hidden at the navel. The top is the root of essence; the navel is the base of life. One root and one base are the origin and ancestors of heaven and earth... It is only these two things, essence and life, that are the truth in thousands of scriptures and myriad discussions" (DZ 244, 2. 10b–11b). He believed that the cultivation of inner nature and life is the foundation of Quanzhen School's Inner Alchemy practice, and identified the head and navel as the locations where inner nature and life reside in the human body. Thus, it can be seen that the early Quanzhen School basically continued the previous Inner Alchemy tradition of equal emphasis on inner nature and life in cultivation.

However, Wang Chongyang and others also advocated that there is a hierarchy and distinction in importance between inner nature and life. Wang Chongyang said, "The root is nature, and life is the base" (DZ 1158, 1a). He also said, "The guest is life, and the host is nature" (DZ 1158, 1b). This expression reveals his inclination to prioritize inner nature over life. To emphasize the importance of nature cultivation, he even said, "inner nature is more important than life" (DZ 1153, 4. 14a). This extreme statement elevates how important it is to cultivate nature rather than life. Ma Yu said, "For humans, there is birth and death, but if there is form, there must be decay" (DZ 1150, 18a). Since the decay of the physical body is an inevitable trend, the significance of nurturing the body is greatly reduced. Hao Datong said, "If a cultivator does not subdue their mind, even if

they have left the mundane world for many years, there is no merit, as they do not see nature of themselves. If they do not see it, how can they nourish life? If inner nature and life are not complete, how can they become true?" (DZ 1256, 1. 20a). To achieve a state of complete inner nature and life, one must start by revealing their true nature and then extend to maintaining the physical body. He also said, "Taking use of life to cultivate inner nature" (DZ 1256, 1. 21a). This explains the hierarchy and importance of the inner nature and life from the perspective of the relationship between them. Qiu Chuji said, "In our sect, the first three stages are all about active cultivation, which is life cultivation; while the latter six stages are the effortless subtle Dao, which is inner nature cultivation. From now on, we only mention nature cultivation instead of life cultivation. The term 'cultivation' implies deliberate efforts. Cultivation is work. With stages and levels, how can nature be considered work? Even Buddha only perfected nature cultivation" (ZW 378, 285). In the Inner Alchemy practice of the Quanzhen School, the proportion of life cultivation was already far smaller than that of nature cultivation, and life cultivation was also set as a deliberate practice, pale in comparison to the inner nature cultivation. However, even this was not satisfactory for Qiu Chuji. He demanded that from now on, only inner nature cultivation should be mentioned, and compared to the Buddhist practice of illuminating the mind and seeing inner nature. This led the Inner Alchemy practice of the Quanzhen School to develop in the extreme direction from canceling life cultivation to focusing on dual cultivation of inner nature and life.

In the early Quanzhen School, while advocating "dual cultivation of inner nature and life", it also demonstrated a tendency to "prioritize inner nature and downplay life" (zhongxing qingming 重性輕命), reflecting the contradiction between secular and religious perspectives on life. For a long time, secular and religious views on life have been intertwined and deeply rooted in traditional Chinese culture, both of which were recognized and accepted by the early Quanzhen School. The secular view of life believes that humans are a physical and spiritual unity, with the latter relying on the former for existence. Based on this concept, the early Quanzhen School inherited the principles and methods of "dual cultivation of inner nature and life" from the Inner Alchemy Daoist tradition, striving to achieve dual liberation of the body and spirit. Fushi practices, as a set of health-preserving methods commonly used in traditional Daoism, belong to the category of life cultivation and were naturally included in the cultivation program by Wang Chongyang and others. On the other hand, the religious view of life holds that physical life can only exist briefly and will eventually fade away, while spiritual life is eternal and indestructible. Therefore, the early Quanzhen School, contrary to the traditional Daoist belief in immortality, sought spiritual liberation and regarded the cultivation of inner nature and the practice of the mind as the main path to becoming an immortal. In this respect, they despised various physical cultivation techniques, including Fushi practices, and focused their attention on the exercise of the mind and nature. It can be said that the contradictory attitude toward Fushi practices in the early Quanzhen School was due to their simultaneous inclusion of two different views on life in ancient China.

## 5. Conclusions

In traditional Taoism, the practice of Fushi has been widely used as a method to maintain health and seek immortality. According to the Quanzhen Taoist documents from the Jin-Yuan period, early Quanzhen Taoism representatives like Wang Chongyang and the 'Seven True Ones' also carried out various Fushi activities. This indicates that most scholars who previously studied Quanzhen Taoism either believed that there was no practice of Fushi within Quanzhen, or thought that their practice of Fushi was influenced by the Southern Lineage of Taoism after the convergence of the northern and southern schools of Taoism. Both of which are inconsistent with the true historical situation of Quanzhen Taoism.

However, as previous researchers pointed out that figures such as Wang Chongyang indeed held a vehemently critical attitude towards Fushi, the root cause of this lies in the

certain level of opposition between the practices of Inner Alchemy and the general health-maintenance techniques, such as Fushi, in terms of the methods of practice and the ultimate goals. Since the late Tang Dynasty and the Five Dynasties, Inner Alchemy emerged from numerous cultivation techniques and held a dominant position. After it absorbed the theoretical components of Confucianism and Zen Buddhism, it gradually emphasized achieving spiritual enlightenment through cultivating the mind and refining the nature. On the other hand, Fushi is a general health-maintenance technique passed down from traditional Taoism, serving the ultimate goal of immortality. Therefore, at the time when Inner Alchemy was absolutely dominant, Fushi was naturally frowned upon and suffered a tremendous impact. As a newly established religious sect at the time, the Quanzhen school urgently needed to demonstrate its superiority and originality in doctrine and practice, hence it negated many traditional Taoist cultivation techniques including Fushi, displaying a confrontational stance against the old religious tradition. The early Quanzhen Taoism's practice of both negating and utilizing Fushi reflects the changes that Fushi underwent in the context of prevailing Inner Alchemy. Looking back at the history of Taoism, Fushi once played a pivotal role in health preservation and cultivation. However, as Inner Alchemy became the mainstream of cultivation and gradually expanded the content of mind–nature cultivation, the existence value of Fushi was greatly weakened. Yet, this does not mean that Fushi then withdrew from the stage of Taoist history. For example, taking drugs, breathing techniques, the emergence of grain avoidance with Inner Alchemy became an auxiliary or even directly acted as an important step in Quanzhen Taoist Inner Alchemy practice. Fushi practices and the use of talisman water were also used to educate the public, becoming part of the Quanzhen Taoist "true practice" (zhenxing 真行) in social life. The Fushi cultivation of early Quanzhen Taoism had a significant impact on the later generations. Among the disciples of the "Seven True Ones", many inherited the Fushi tradition of early Quanzhen Taoism. Yin Zhiping 尹志平 (1169–1251) believed that the quality of daily diet should match the depth of cultivation, thus urging himself to accumulate meritorious deeds. At the same time, he drank tea to assist in religious cultivation against sleepiness. Yang Mingzhen 楊明真 (1154–1233) and Fan Yuanxi 范圓曦 (1178–1249) both gained renown by using talisman water to cure patients. Wang Zhijin 王志謹 (1178–1263) repeatedly emphasized temperance in eating and drinking and rejection of alcohol and meat, which is consistent with the dietary principles of early Quanzhen Taoism. Shi Chuhou 史處厚 (1101–1174) tried "abstaining from grains" and "only drinking Daogui", and Li Zhichang 李志常 (1193–1256) also had experience with "grain avoidance for several weeks". Obviously, these disciples' Fushi practices directly inherited the cultivation methods of early Quanzhen Taoism. Unlike the previous generation of Quanzhen Taoist masters, they hardly expressed any disdain or dissatisfaction with Fushi in their writings. This indicates that by that time, Fushi had completely integrated into the daily preaching and cultivation activities of Quanzhen Taoism, and hence the inherent conflict between the old Taoist tradition and the new religion and its new cultivation method also dissolved.

The active use of Fushi practices by the early Quanzhen School contributed to the rapid development and expansion of its sect. First, Fushi practices enriched the cultivation methods of the Quanzhen School. In the teachings of Zhong-Lü, there was already a combination of Inner Alchemy cultivation and Outer Alchemy Fushi practices.[11] The Quanzhen School further incorporated breathing exercises, abstaining from grains, and eating and drinking into the Inner Alchemy perspective, making them part of the Quanzhen School's Inner Alchemy techniques, which promoted the perfection of the Quanzhen School's cultivation system. Second, Fushi practices were often used as a medium to connect Quanzhen practitioners with the general public, building a solid foundation of support for the Quanzhen School. In order to attract and educate as many people as possible and expand the influence of the religious group, the early Quanzhen School frequently made use of Alchemy techniques, including Fushi techniques. Wang Chongyang and others interacted with the public through healing methods such as talisman water and giving medicinal

remedies, as well as imparting health preservation knowledge in daily eating and drinking practices, adhering to the principles of asceticism and inner nature and life cultivation. This not only created a positive image for the religious group but also subtly instilled the sacred teachings of the sect into the general public, creating favorable conditions for a large number of people to learn about and join the Quanzhen School. In conclusion, the rapid rise of the Quanzhen School during the Jin and Yuan periods was largely attributed to the active engagement in Fushi practices.

**Author Contributions:** Conceptualization H.C.; writing—original draft preparation, H.C.; writing—review and editing, Y.H. All authors have read and agreed to the published version of the manuscript.

**Funding:** This research was funded by The National Social Science Fund of China, grant number 21AZJ005.

**Institutional Review Board Statement:** Not applicable.

**Informed Consent Statement:** Not applicable.

**Data Availability Statement:** Not applicable.

**Conflicts of Interest:** The authors declare no conflict of interest.

## Notes

[1] Ishijima Yasutaka石島快隆 (Ishijima 1960) believes that the ideas from *Token for the Agreement of the Three According to the Book of Changes* (Zhouyi cantong qi 周易參同契) by Wei Boyang 魏伯陽 (fl. circa 132–168), a pioneering Daoist of cultivation and Fushi practice, and systematized *Book of the Master Who Embraces Simplicity* (Baopu zi 抱朴子) by Ge Hong 葛洪 (283–363) are the pillar of Yijing studies and the Daoist thoughts of immortals influenced by Yin-Yang Five Elements (wuxing 五行) during Qin (221–207 BC) and Han (202 BC–220 AD) periods. Yan Jinxiong's 顏進雄 (Yan 2000) *The Influence of Fushi Customs and Poetry in the Six Dynasties* (Liuchao fushi fengqi yu shige 六朝服食風氣與詩歌) examines the background, thought, and types of Fushi during the Six Dynasties (222–589) period, with a focus on the influence of Fushi on the development of poetry during this time. Liao Ruiyin's 廖芮茵 (Liao 2004) *Study of Fushi for Health Preservation in the Tang Dynasty* (Tangdai fushi yangsheng yanjiu 唐代服食養生研究) discusses the era environment of Fushi in the Tang Dynasty, sorting out the practices of emperors, nobles, literati, Daoists, and Buddhists, and provides a rational evaluation of the contributions and influences of Fushi on health preservation in the Tang Dynasty. Huang Yongfeng's 黃永鋒 (Huang 2008) *Research on Daoist Fushi Techniques* (Daojiao fushi jishu yanjiu 道教服食技術研究) redefines the scope of Daoist Fushi, using the framework of the philosophy of technology, and explores its procedural characteristics and practical functions, revealing the driving mechanism of its development. Xu Gang's 徐剛 (Xu 2018) *Daoist Fushi Research from the Perspective of Life Philosophy* (Shengming zhexue shiyuxia de daojiao fushi yanjiu 生命哲學視域下的道教服食研究) analyzes the relationships between Daoist physiology, philosophy, body–spirit, change, and Fushi from the perspective of life philosophy, and uses modern scientific statistical analysis to investigate the ingredients, techniques, measurements, prescriptions, and nutritional components of Daoist Fushi recipes. *Fushi* by He Zhenzhong何振中 (He 2022) organizes its history from the Qin (221–207 BC) and Han Dynasties (202 BC–220 AD) to the modern era, introduces the types, efficacy, preparation methods and uses, explores the relationship between Fushi and inner refinement (neilian 內煉), guiding and pulling (daoyin 導引), grain avoidance and other health preservation techniques and reveals how significant and valuable to inherit the traditional Fushi. Some progress has also been made in the collation of Fushi literature. Chen Guofu 陳國符 (Chen 2014, pp. 378–401) and Jiang Lisheng 蔣力生 (Jiang 2004a, pp. 21–23; 2004b, pp. 18–23) have both compiled bibliographies focusing on ancient Fushi literature. Li Ling 李零 (Li 1993), the chief editor of *An Overview of Chinese Alchemy* (Zhongguo fangshu gaiguan 中國方術概觀), has proofread and collated eight Fushi texts from the *Zhengtong Daoist Canon* (Zhengtong daozang 正統道藏), with summaries written before each book, introducing the content and overall value. There are many other monographs and numerous papers discussing Fushi from different perspectives, which are too numerous to mention here.

[2] Huang Yongfeng believes that Daosim Fushi includes five types of taking herbal medicine, breathing exercises (qi), abstaining from grains, eating and drinking, and using talismans. He discussed each of them (Huang 2008, pp. 100–69), which is refereed by the author.

[3] Liu Xianxin 劉咸炘 (1896–1932) believes that northern and southern lineages of the Quanzhen "never mention Fushi practices" (Liu 2012, p. 46). Hou Guangfu 侯光復 says, "As a matter of fact, the Quanzhen Daosim really look down on traditional Daoist alchemical arts of cinnabar and mercury . . . Other well-known Quanzhen masters also do not mention any techniques for Fushi practices" (Hou 1988, p. 90). Zhu Zhanyan朱展炎 states that "cultivation and Fushi practices were strongly criticized by the early Quanzhen predecessors", and believes that "Quanzhen negates traditional cultivation practices which are fundamentally on the basis of its pursuit of the theoretical foundation of mental liberation" (Zhu 2009, pp. 96–97). Xiao Jinming 蕭進銘 also asserts, "That Quanzhen does not mention Fushi practices is true" (Xiao 2010, p. 674).

4   *Inner Chapters of Master Who Embraces Simplicity* (Baopu zi neipian 抱朴子內篇) writes that there are five types of zhi 芝, including stone zhi, wood zhi (muzhi 木芝), plant zhi (caozhi 草芝), flesh zhi (rouzhi 肉芝), and mushroom zhi (junzhi 菌芝). Each of these five types have more than a hundred of kinds of it (See DZ 1185).

5   Meng Naichang 孟乃昌 (1933–1992) believes that Qiu Chuji knew external elixir and valued it. He provides the following evidence: In Qiu Chuji's *Poems on Spring Outings during Cold Food Festival* (Hanshi ri zuo chunyou shi 寒食日作春遊詩), a line reads: "I wish I could acquire the great elixir to strengthen body to fly up to the heaven". This reflects his aspiration and unfulfilled hope for success in external elixir. Ye Ziqi 葉子奇 (fl. circa 1327–1390) of the Ming Dynasty recorded in his *Master of Grass and Trees* (Caomuzi 草木子) that: "Qiu Chuji (also known as Changchun zi 長春子) was able to refine cinnabar into gold, thereby providing financial support for the military and national affairs of the Yuan Dynasty. Due to this achievement, he was awarded a golden seal and became the leader of the Quanzhen Taoism". While the claim that Qiu Chuji helped finance Kublai Khan's 忽必烈 (1215–1294) military by refining gold and silver is not reliable, it may be related to his experiments with refining during his lifetime. Additionally, the White Clouds Temple 白雲觀, where Qiu Chuji stayed and preached after returning from the western regions to Yanjing, also has a tradition of practicing external elixir (Meng 2018, p. 124).

6   Abstaining from grains consists of two types, the natural one and intentional one. The former refers to when a person reaches a certain level of cultivation, their qi and blood are in abundance, and they do not desire to consume grains. The latter refers to deliberately not consuming grains or food during cultivation, and instead eating other fruits, herbs, etc.

7   Chen Yingning's 陳攖寧 (1881–1969) *Commentary to the Practice stages of Kundao* (Sun Bu'er nügong neidan cidi shi zhu 孫不二女功內丹次第诗注) states: "This is the true practice of abstaining from food. Those who can achieve this do so because the spiritual energy is full within their bodies and they naturally do not think about food. This is not because they can endure hunger with an empty stomach" (Hu and Wu 2008, p. 90). *Explication of the Practice stages of Kundao* (Sun Bu'er yuanjun kundao gongfu cidi zhushi 孫不二元君坤道功夫次第注释) says: "During Greater Celestial Circuit (da zhoutian 大周天), one consumes two types of qi, and the desire for food is cut off. This is what we call the abstaining from grains (bigu 辟穀). From this, it can be seen that abstaining from grains is just a natural outcome" (Chen 1975, p. 9). Both of these statements indicate that abstaining from grains is not a proactive action, but an effect that occurs after female elixir practitioners reach a certain level. This is what we commonly refer to today as, "Full qi within leads to the cutting-off of food".

8   Qiu Chuji believes that spring belongs to the wood element in the Five Elements, corresponding to the liver in the five internal organs and sourness in the five flavors. According to the theory of mutual restraint among the Five Elements, wood overcomes earth, and earth corresponds to the spleen in the five internal organs and sweetness in the five flavors. Therefore, the dietary principle for spring is "appropriate reduction of sourness and increase of sweetness to nourish the spleen's qi". Following this, in summer, it is "appropriate to reduce bitterness and increase spiciness to nourish lung qi"; in autumn, it is "appropriate to reduce spiciness and increase sourness to nourish liver qi"; and in winter, it is "appropriate to reduce sourness and increase bitterness to nourish heart qi". As for the dietary dos and do nots in the four seasons, Qiu Chuji has more detailed rules, such as not drinking excessively in spring, and elderly people should not eat too much rice, noodles, or pastries. In summer, one should not eat cold or greasy food, nor melons, eggplants, or uncooked vegetables. Instead, one should drink cassia soup and cardamom boiled water, and eat more beans. In autumn, one can eat more sesame; in winter, avoid eating too much barbecue, meat noodles, and wontons. See Qiu (2005, pp. 96–105).

9   During the Tang and Song (960–1279) dynasties, the alchemy of Daosim transformed from the external to the internal. At that time, a transitional stage, alchemy practice leading to the Dao, emerged in internal alchemy. Earlier scholars pay relatively less attention to it. Guo Wu郭武 believes that the aim of "alchemy practice leading to the Dao" was to criticize the old to establish the authority of the new. This phenomenon had positive significance in the late Tang Dynasty shortly after the rise of internal alchemy practice (Guo 2016, p. 54).

10  Zhongli Quan said: "There are three types of illnesses. Seasonal illnesses are cured by taking herbal medicine. Physical and geriatric diseases often need to be treated by taking two types of medicine, namely internal and external elixir" (DZ 263, 15. 8a–b). External elixir is not unusable. He also stated: "It is just because people who practice the Dao may only awaken in their later years and are aware of the fact that their kidney and heart are not healthy. The kidney is like the root of qi. If the root is not deep enough, the tree's leaves will not grow abundantly. The heart is like the source of fluids. If the source is not crystal, the water will not flow continuously. It is necessary take a long time to refine the Dragon-Tiger Great Elixir with Nine Grades (jiupin longhu dadan 九品龍虎大丹) by using various minerals. This can helps the practitioner to connect with the true qi and refine their body so as to allow it to stay in the world immortally, and ultimately ascend to heaven" (DZ 263, 15. 10b). It can be seen that Zhongli Quan and Lü Dongbin not only did not oppose taking medicine and elixirs, but even affirmed their effects.

11  *A True Record of the Assembled Immortals of the Western Hills* (Xishan qunxian huizhen ji 西山群仙會真記) states "From ancient times to the present, wise men have also discussed external alchemy, so it is not that external alchemy cannot be used. . . . From this, we can see that after the spirit goes through gathering and dispersing changes, it ultimately becomes void. Using qi to return to the origin is called returning alchemy. Later generations of people who took external alchemy also saw its effects and became immortals. This is because they began practicing external alchemy but also practiced internal alchemy at the same time. Both internal and external alchemy achieved results, so they understood the way to become immortals. If one only uses external alchemy, their qi will weaken, their spirit will deteriorate, and the refined qi of heaven and earth cannot condense within their body, resulting in significant harm instead" (DZ 246, 4. 7a–b). This shows that although the Zhong-Lü school advocates internal

alchemy, they do not reject external alchemy. In their view, using internal alchemy alone can make one an immortal, while external alchemy must be combined with internal alchemy to achieve immortality.

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
