# Peer review of "A New Study on Fushi of Early Quanzhen Daoism"

_religions, doi:10.3390/rel14060814_

Round 1

Reviewer 1 Report

The topic of the article is interesting in the field to which it belongs. The structure is quite well managed, but needs improvements such as: 1) The conclusions are weak: they serve to pull the strings of what was said in the article. It is best to avoid too many quotations and outline the contributions of concepts mentioned above. 2) it is better to report some high-level specialist studies to support the hypothesis: even if there are no specific contributions in the literature, there are texts that can frame the hypothesis 3) the abstract: a conclusion is needed. 4) on line 67 it is better to insert "interesting" instead of "essential".

Author Response

1)The paper has been revised.

2)This article focuses on the relationship between Quanzhen Taoism and the Taoist practice of Fushi. However, scholars who have previously studied Quanzhen Taoism have rarely paid attention to Fushi within Quanzhen. Even scholars who have paid attention to this aspect have directly determined that there is no Fushi in Quanzhen based on a few critical remarks about the practice by Quanzhen. And scholars studying Taoist Fushi conduct their research based on Taoism as a whole without examining the specific situation of different sects. Therefore, to date, no scholar has systematically studied the relationship between Quanzhen Taoism and Fushi.

In our research, we primarily referenced Huang Yongfeng's黃永鋒 Research on Daoist Fushi Techniques (Daojiao fushi jishu yanjiu道教服食技術研究) and Zhu Zhanyan's朱展炎 Taming the Self - A Study on Wang Changyue's Taoist Thoughts (Xunfu ziwo - Wang Changyue xiudao sixiang yanjiu馴服自我——王常月修道思想研究). The former divides Taoist practice of Fushi into five types: taking medicine (fuyao服藥), breathing exercises (fuqi服氣), fasting (bigu辟穀), dieting (yinshi飲食), and using talismans (fufu服符). This classification provides us with clues to investigate the Fushi practices within Quanzhen Taoism. The latter acknowledges the practice of Fushi in Quanzhen Taoism. His reason is that according to Wang Hui's王褘 description, Fushi is one of the notable features of Quanzhen Taoism at the end of the Yuan Dynasty. He recognizes the reliability of Wang Yi's records and also notes that this record differs from the impression that Quanzhen Taoism does not involve Fushi and considers this to be a result of the convergence of the northern and southern Taoist sects during the middle of the Yuan Dynasty. However, according to our research, there were numerous instances of Fushi in Quanzhen Taoism at its inception, and this was not the result of the convergence of Taoist sects. Therefore, we wrote this article with a focus on early Quanzhen Taoism.

3)The paper has been revised.

4)The paper has been revised.

Reviewer 2 Report

This manuscript explores the relationship between Fushi, a body cultivation technique highly valued and widely used in traditional Daoism, and early Quanzhen Daoism, founded by Wang Chongyang during the Jin Dynasty. The manuscript focuses on Wang Chongyang and the Qizhen to study their relationship with Fushi and investigate the contradiction between words and pratiques in early Quanzhen Daoism related to internal alchemy thinking. The manuscript suggests that the contradictions observed among the early Quanzhen Daoists are rooted in the internal alchemy mindset, which prioritizes dao over shu and xing over ming.

The manuscript effectively tackles the relationship and conflict between the early Quanzhen Daoists' rejection of traditional Fushi techniques and their extensive engagement in similar complementary activities. The author diligently supports their perspective with multiple references. Exploring this relationship is crucial for a precise understanding of the inheritance dynamics between Quanzhen and traditional Daoist techniques.

This manuscript presents an insightful question and provides an informative, well-documented discourse in response. However, the manuscript may be difficult to understand for those with limited knowledge of Daoism, specifically the internal alchemy practices, and the historical context of Quanzhen. Furthermore, we look forward to seeing the author engage in dialogue with Western scholars researching Quanzhen Daoism within this manuscript.

Here are some specific remarks:

Line 24-55

The manuscript provides a thorough review of previous literature on the subject of Fushi, but it is important to note that a literature review should not merely consist of a listing of studies. Instead, it should highlight the gaps or shortcomings in prior research and articulate how the present paper can contribute to the academic discourse. A more efficient approach would involve directly addressing the viewpoints of past scholars regarding the relationship between the Quanzhen sect and Fushi in the Introduction section.

Line 67-60

The author presents a highly insightful academic question. However, they have not elucidated the new materials they will employ to address this question. It is crucial for the author to provide a succinct introduction to the materials they will utilize within the Introduction section.

Line 60-68

The manuscript does not clarify why the focus is on the Quanzhen sect. In the first paragraph, there is no mention of the sect's association with Fushi practices. Additionally, when discussing the research findings of other scholars, there is no mention of whether they have explored other Daoist sects such as Shangqing or Zhengyi, which may address the ideas and practices of Fushi. If the author intends to emphasize the uniqueness of the Quanzhen sect in the second paragraph, it would be appropriate to provide a brief overview of other sects' perspectives before that.

Furthermore, it would be beneficial for the author to provide a concise explanation of the Quanzhen sect, along with relevant notes and references. The usage of certain terms in the sentence is ambiguous, such as the meaning of the sect being "newly" established. It is necessary for the author to provide a clearer definition of the Quanzhen sect for the readers. Also, what other techniques did Quanzhen adopt a rebellious attitude towards in their opposition to traditional Daoist cultivation techniques?

Line 82

What academic community? 

Same as in lines 60-68, the author needs to provide notes.

Line 84-90

The author needs to provide the original text, not just the translation. 

The author should introduce Wang Yi and Qingyan Conglu. From what standpoint did Wang Yi make this statement? Is this statement representative?

Line 153

Prior to delving into the discussion, it is imperative for the author to provide a preliminary explanation of how the structure of Chapter 3 revolves around the description of these five practices.

Line 197-199

The author should be more careful when referencing the works of Sun Bu'er, as these works could potentially be creations from the Qing Dynasty.

Line 387

It is necessary for the author to provide a clearer definition of Dao and Shu. Specifically, it should be clarified that the corresponding Chinese character for "technique" is "shu," rather than introducing this information later in line 439.

Line 440

Is the Zhong-Lü Daoist school an authentic historical entity, or is it a later construction by Daoist masters?

Line 596

“The Fushi of the early Quanzhen School had a significant impact on later generations.”

In this section, the author only references the viewpoints of Daoist masters from the 12th and 13th centuries. what about the impact on later generations? Did they start to question the Fushi practices of early Quanzhen Daoists? Does this correspond to the discussion in Chapter 2?

Line 623-642

This conclusion is highly insightful and thought-provoking. Could the author provide additional evidence to further substantiate the claim that Fushi practices contribute to the development of Quanzhen?

Syntax and words should be improved.

Author Response

Line 24-25

The paper has been revised.

Line 67-60

Relevant explanations have been supplemented in the paper.

Line60-68

Since Taoism incorporated Fushi practices into its cultivation system, it has developed a rich variety of Fushi programs. Previous scholars have summarized Taoist Fushi practices into five types: taking medicine, absorbing qi, grain avoidance, diet, and taking talisman mater, which basically cover all of Taoist Fushi activities. Therefore, we will start from these five aspects to examine the Fushi practices of early Quanzhen Taoism. Previous scholars studying Taoist Fushi practices have tended to consider Taoism as a whole. They haven’t distinguished the individual situations of different sects. It has become a consensus among Taoist scholars that traditional Taoism recognizes and utilizes Fushi practices. Quanzhen Taoism emerged relatively late and was initially just a community of practitioners. Until the third year 1198 in the reign of Emperor Zhangzong of the Jin Dynasty (1115-1234), when the emperor granted the name Lingxu Guan靈虛觀 to the hermitage of Quanzhen Taoism in Zhongnan Mountain, Quanzhen Taoism officially embarked on the ancient Taoist tradition. Fushi, with their long history and strong influence, are highly representative in the many health-cultivation techniques of traditional Taoism. Whether they would be recognized and utilized by Quanzhen Taoism, a newly established Taoist sect, is a question worth studying. Previous scholars have not specifically researched the relationship between Quanzhen Taoism and Fushi practices. In addition to Fushi, Quanzhen taoism also opposed techniques such as swallowing saliva (yanjin嚥津), visualization and imagination (cunxiang存想), seated meditation (dazuo打坐), arts of the bedchamber (fangzhongshu房中術), etc.

Line 82

The original text has been deleted.

Line84-90

The paper has been revised.

Line153

Additional explanations have been provided.

Line197-199

At present, the works related to Sun Bu'er孫不二 include Dharma Sayings of primordial goddess Sun Buer (Sun Bu’er yuanjun fayu孫不二元君法語), Secret writings on the way of the elixir transmitted by primordial goddess Sun Bu'er (Sun Bu'er yuanjun chuanshu dandao mishu孫不二元君傳述丹道秘書), Instructions of the Female (Kunjue坤訣), Scripture of the Original Female by primordial goddess Qingjing (Qingjing yuanjun kunyuan jing清靜元君坤元經), Ten Principles of the Queen Mother of the West on the Correct Path of Female Cultivation (Xiwangmu nüxiu zhengtu shize西王母女修正途十則), etc. Additionally, a few poems signed by Sun Bu'er are included in the poetry collection Echoes of Cranes’ Songs (Minghe yuyin鳴鶴餘音). Among them, Sun Bu’er yuanjun fayu孫不二元君法語 is generally considered to be Sun Bu'er's work and is highly credible. Other works are either hard to be distinguished as real ones or are ascribed to Sun Bu'er.

Line387

The paper has been revised.

Line440

The so-called Zhong-Lü Golden Elixir School鍾呂金丹派, centered around the legend and belief of Zhongli Quan鍾離權 and Lü Dongbin呂洞賓, was interpreted and developed by Inner Alchemy practitioners from the Five Dynasties to the Northern Song Dynasty. It is just a synonym for the Inner Alchemy Taoism after the Five Dynasties, not an actual sect. Anthology of Zhongli Quan’s Transmission of the Dao to Lü Dongbin (Zhong-Lü chuandao ji鍾呂傳道集) records the discourse on Inner Alchemy cultivation by Zhongli Quan and Lü Dongbin, and is the most systematic Inner Alchemy work between the Tang and Song Dynasties, serving as the doctrinal source of the Zhong-Lü Golden Elixir School. This book has been widely circulated in society from the beginning of the Northern Song Dynasty to the Southern Song Dynasty. Wang Chongyang王重陽 said that Zhongli Quan was his grandmaster, Lü Dongbin was his master, and Liu Haichan劉海蟾 (a Taoist from the Five Dynasties who was taught the Inner Alchemy by Zhongli Quan and Lü Dongbin) was his uncle. This should reflect that Wang Chongyang was influenced by Zhong-Lü Inner Alchemy School in terms of practicing Inner Alchemy. Therefore, the insights of Wang Chongyang and his disciples on religious cultivation, especially Inner Alchemy practice, can be traced back to the Zhong-Lü School.

Line596

The paper has been revised.

Line623-642

The direct evidence of Fushi assisting in the early Quanzhen Taoist alchemy practice is detailed in my article. Additionally, Fushi has contributed to the early Quanzhen Taoism to unite the public. According to the records of The Quanzhen Taoism Ancestor Stele of Immortal Chongyang in Zhongnan Mountain (Zhongnan shan shenxian Chongyangzi Wang zhenren Quanzhen jiaozu bei終南山神仙重陽真人全真教祖碑), when Wang Chongyang王重陽 was preaching in Shandong, he created many miraculous events which won the local people over and they willingly submitted to him. Zhongnan Mountain Chongyang Ancestor Immortal Trace Record (Zhongnan Shan Chongyang. Zushi xianji ji終南山重陽祖師仙跡記) also recorded many of Wang Chongyang's methods and techniques which was referred to as “Expedient Cognition (quanzhi權智)” which are temporary means used to lead people to achieve real cognition (shizhi實智). Yin Zhiping尹志平, in the Recorded Sayings from a Journey to the North (Qinghe zhenren beiyou yulu清和真人北遊語錄) recalled his master Qiu Chuji's丘處機 experience of teaching and also believed that the implementation of teaching needed to use expedient cognition to educate people. From this, it can be seen that both Wang Chongyang and Qiu Chuji used technical arts extensively to rally the public. Although these materials do not directly mention Fushi, it also belongs to the technical arts they utilized, and therefore can indirectly reflect its functions such as gathering believers, strengthening the organization and disseminating doctrine."